# OpenReview forum: "AHELM: A Holistic Evaluation of Audio-Language Models"
_ICLR.cc/2026/Conference — ICLR 2026 Conference Withdrawn Submission_

### Official Review · Reviewer_qDSB · 2025-10-31

**Soundness:** 3
**Presentation:** 3
**Contribution:** 3
**Rating:** 6
**Confidence:** 4

**Summary:**

This paper introduce AHELM, a holistic benchmark for audio-language models (ALMs). The benchmark assesses model performance across ten key dimensions: audio perception, knowledge, reasoning, emotion detection, bias, fairness, multilinguality, robustness, toxicity, and safety. The authors evaluate 14 systems, including both open-source and closed-API models, uncovering previously unreported weaknesses and providing valuable insights for future model development.

**Strengths:**

* The primary contribution of this work lies in integrating previously underexplored evaluation dimensions for audio-language models (bias, fairness, robustness, toxicity and safety) into a unified assessment framework. This holistic design broadens the scope of evaluation beyond traditional perception and reasoning metrics.
* The introduction of two new datasets, CoRE-Bench (multi-turn conversational audio reasoning) and PARADE (stereotype probing), fills critical gaps in existing benchmarks.
* The discussions are well-reasoned and provide a solid understanding of the broader challenges of current audio-language models.

**Weaknesses:**

* The paper lacks sufficient discussion of related work and omits direct comparisons with existing audio-language model benchmarks such as AIR-Bench[1], MMAU[2], Dynamic-SUPERB[3,4], and VoiceBench [5], which already explore evaluation in multi-aspect overlapped with this work. A comparative analysis against these frameworks would help clarify AHELM’s distinct contributions and positioning.
* The originality of the benchmark is somewhat limited, as most evaluation tasks are directly adopted from existing datasets rather than newly designed or substantially modified.
* The coverage and completeness of certain evaluation aspects remain constrained. For instance, in the bias and fairness dimension, only gender is considered, leaving out other critical attributes such as accent, age, and ethnicity. Similarly, in multilinguality evaluation, only two language pairs are included for speech translation, which restricts the generalizability of the findings. Lastly, robustness focuses only on additive noise without considering reverberation, clipping, etc.
* [Minor] From a “holistic evaluation” perspective, it would be beneficial to include a broader range of ALMs. Currently, the open-source category includes only the Qwen-family models. Incorporating other systems, such as the Audio-Flamingo series, and older model like SALMONN and WavLLM, would yield a more representative comparison and provide valuable insights into the progress and evolution of the community’s efforts over time.

[1] Yang, Qian, et al. "AIR-Bench: Benchmarking Large Audio-Language Models via Generative Comprehension." ACL 2024. 2024.

[2] Sakshi, S., et al. "MMAU: A Massive Multi-Task Audio Understanding and Reasoning Benchmark." ICLR 2025. 2025.

[3] Huang, Chien-yu, et al. "Dynamic-superb: Towards a dynamic, collaborative, and comprehensive instruction-tuning benchmark for speech." ICASSP 2024, 2024.

[4] Huang, Chien-yu, et al. "Dynamic-SUPERB Phase-2: A Collaboratively Expanding Benchmark for Measuring the Capabilities of Spoken Language Models with 180 Tasks." ICLR 2025. 2025.

[5] Chen, Yiming, et al. "Voicebench: Benchmarking llm-based voice assistants." 2025.

**Questions:**

* Why are aspects such as hallucination detection, instruction-following ability not included? These capabilities are increasingly important for evaluating multimodal large language models [6,7] and would further broaden the coverage of this benchmark.
* The reliance on mean win rate (MWR) as the primary aggregation metric may limit benchmark stability and reproducibility, since results can shift as new models or datasets are added. How do the authors plan to mitigate this issue to ensure long-term comparability and transparency for benchmark users?

[6] Kuan, Chun-Yi, et al. "Understanding Sounds, Missing the Questions: The Challenge of Object Hallucination in Large Audio-Language Models." Interspeech 2024. 2024.

[7] Gao, Yiming, et al. "IFEval-Audio: Benchmarking Instruction-Following Capability in Audio-based Large Language Models." 2025.

---

### Official Review · Reviewer_STuJ · 2025-10-31

**Soundness:** 2
**Presentation:** 2
**Contribution:** 1
**Rating:** 2
**Confidence:** 5

**Summary:**

This paper introduces AHELM, a benchmark aiming to comprehensively evaluate audio-language models (ALMs). The proposed benchmark covers 10 aspects of ALMs by aggregating existing benchmarks and two newly proposed datasets. The evaluation reveals limitations of current ALMs.

**Strengths:**

- Introducing two new datasets.
- The paper writing is clear enough.

**Weaknesses:**

### Missing References
While the authors claim that AHELM provides a holistic evaluation framework, it overlooks several important aspects and lacks sufficient references to prior work. For instance, multiple benchmarks have already been proposed to assess different dimensions of reasoning in ALMs [1–4]. Although these benchmarks target different aspects compared to the proposed CoRe-Bench, it remains essential for the authors to acknowledge and cite these works to improve the completeness and contextual grounding of the paper.

Furthermore, the current version of the paper may give readers the impression that the proposed PARADE benchmark is the first to evaluate bias and stereotypes in ALMs. This framing is inaccurate, as several previous studies have already explored bias and stereotyping in ALMs [5,6]. The authors are strongly encouraged to revise the related work section to properly acknowledge these contributions and clarify how PARADE differs from or extends them.

Finally, to strengthen the coverage and accuracy of references, the authors may refer to a recent survey on ALM evaluation [7], which provides a comprehensive overview of related efforts. Overall, I believe the current version of the paper insufficiently recognizes prior research, which weakens its scholarly positioning.



[1] Sakshi et al., "MMAU: A Massive Multi-Task Audio Understanding and Reasoning Benchmark", ICLR 2025

[2] Yang et al., "SAKURA: On the Multi-hop Reasoning of Large Audio-Language Models Based on Speech and Audio Information", Interspeech 2025

[3] Ma et al., "MMAR: A Challenging Benchmark for Deep Reasoning in Speech, Audio, Music, and Their Mix", arXiv preprint

[4] Yang et al., "SpeechR: A Benchmark for Speech Reasoning in Large Audio-Language Models", arXiv preprint

[5] Lin et al., "Listen and Speak Fairly: a Study on Semantic Gender Bias in Speech Integrated Large Language Models", SLT 2024

[6] Lin et al., "Spoken Stereoset: on Evaluating Social Bias Toward Speaker in Speech Large Language Models", SLT 2024

[7] Yang et al., "Towards Holistic Evaluation of Large Audio-Language Models: A Comprehensive Survey", EMNLP 2025

### Coverage of ALMs

The current coverage of ALMs is quite restricted. I believe it will be much better if the authors can report performances of more ALMs.

### Experimental Results

I would suggest the authors to move the results of ALMs on the CoRe-Bench and PARADE to the main text of the paper. In addition, please consider to include statistical measures to demonstrate the significance of the reported results.

In addition, since the proposed benchmark highly relies on existing ones, the insights is not quite novel, which will undermine the contribution of this paper.

**Questions:**

I have listed my concerns in the weakness section.

---

### Official Review · Reviewer_4EZE · 2025-10-31

**Soundness:** 1
**Presentation:** 2
**Contribution:** 2
**Rating:** 2
**Confidence:** 4

**Summary:**

The paper presents AHELM, a benchmark designed for evaluating audio-language models (ALMs) across ten dimensions: audio perception, knowledge, reasoning, emotion detection, bias, fairness, multilinguality, robustness, toxicity, and safety. It assesses 14 ALMs and three ASR+LM baselines. Experimental results show that Gemini 2.5 Pro achieves the best overall performance but exhibits unfairness in certain settings. Additionally, the authors also found that simple ASR+LM pipelines perform competitively in several areas, particularly in robustness and certain emotion recognition tasks.

**Strengths:**

- The benchmark covers ten clear-defined aspects that cover a wide range of speech and audio tasks drawn from existing datasets and benchmarks. In addition, the authors curated two new datasets to address specific aspects not covered by existing resources.
- The authors standardized the evaluation setup by using fixed zero-shot prompts and consistent decoding constraints, which helps reduce variance across different models.
- The plan to release prompts, generations, and a leaderboard is aligned with reproducibility and community adoption.

**Weaknesses:**

- In the abstract, the authors claim there is a lack of standardized ALM benchmarks and that existing ones cover only one or two aspects. This is doubtful. Benchmarks such as AIR-Bench and Dynamic-SUPERB already cover more than a few aspects.
- As a benchmark paper, it does not review prior work on ALM benchmarks. The related work section only discusses ASR benchmarks.
- The framework highlights "adaptation" as a key part, but all experiments are zero-shot. The reason to include adaptation in the framework is unclear.
- The ten aspects are clearly defined, but the rationale is not. Why do these ten yield a holistic evaluation? Is there prior work or information that supports this grouping?
- The 14 ALMs tested are mainly Gemini, GPT, and Qwen, and Qwen is the only open-source model. The paper says it selects "popular state-of-the-art" models for a fair comparison, which I do not consider an appropriate statement. The criteria for "state of the art" are not stated, and several open-source models from the community are missing.
- Many findings are not new. For example, the first two points were already observed in Dynamic-SUPERB Phase-2. For the emotion result, note that MELD extends EmotionLines, a text-only sentiment dataset, so its spoken content naturally carries cues for emotion.
- The use of an LLM as a judge in this paper is questionable. The reported agreement rate is only around 50%, which makes the LLM-based scores unreliable. Moreover, using GPT-4o as the evaluator raises reproducibility concerns. Even with the temperature fixed at 0, GPT-4o outputs can vary due to hidden randomness or version updates. Since OpenAI frequently modifies its models, it is unclear how long the version used in this paper will remain accessible. Although the authors attempt to fix the inference configuration, the evaluation setup is non-reproducible.

**Questions:**

Please refer to my comments in the weakness section. I also have a few minor comments:
- Figure 1 is never mentioned in the main text. It appears only once in the caption of Table 1, and it is unclear whether this is intentional.
- There are several typos in the paper, such as "Samping Rate" in Table A3, "unaswerable" in Section E.4.4, and "µmale and µmale" in Appendix H.
- Figure 4 is difficult to read because it includes too many models with limited colors, and the data points for different models overlap heavily.

**Details Of Ethics Concerns:**

Since AHELM includes data used to evaluate bias, fairness, toxicity, and safety, I believe the paper should undergo an additional ethics review to ensure that potential ethical concerns are properly addressed.

---

### Note · Authors · 2025-11-17

I have read and agree with the venue's withdrawal policy on behalf of myself and my co-authors.